

# Unfrustrating the $t$-$J$ model:
# d-wave BCS superconductivity in the $t'$-$J_z$-$V$ model

**Kevin Slagle**[1,2⋆]

**1** Walter Burke Institute for Theoretical Physics,
California Institute of Technology, Pasadena, California 91125, USA
**2** Institute for Quantum Information and Matter,
California Institute of Technology, Pasadena, California 91125, USA

⋆ kslagle@caltech.edu

## Abstract

The $t$-$J$ model is believed to be a minimal model that may be capable of describing the low-energy physics of the cuprate superconductors. However, although the $t$-$J$ model is simple in appearance, obtaining a detailed understanding of its phase diagram has proved to be challenging. We are therefore motivated to study modifications to the $t$-$J$ model such that its phase diagram and mechanism for d-wave superconductivity can be understood analytically without making uncontrolled approximations. The modified model we consider is a $t'$-$J_z$-$V$ model on a square lattice, which has a second-nearest-neighbor hopping $t'$ (instead of a nearest-neighbor hopping $t$), an Ising (instead of Heisenberg) antiferromagnetic coupling $J_z$, and a nearest-neighbor repulsion $V$. In a certain strongly interacting limit, the ground state is an antiferromagnetic superconductor that can be described exactly by a Hamiltonian where the only interaction is a nearest-neighbor attraction. BCS theory can then be applied with arbitrary analytical control, from which nodeless d-wave or s-wave superconductivity can result

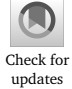

# 1   Introduction

The $t$-$J$ and Hubbard models have been studied extensively as toy models for high-temperature superconductivity in the cuprate superconductors [1–4]. However, the ground states of these models and materials are often frustrated by multiple competing or intertwining orders [5]. For example, in the $t$-$J$ model, the antiferromagnetic Heisenerg term $J$ results in antiferromagnetic order at half-filling; however, when the system is hole doped, then the hopping of holes will locally destroy the antiferromagnetic alignment. The competition between the $t$ and $J$ terms makes well-controlled analytical study of the $t$-$J$ model difficult.

Nevertheless, one might hope to find a corner of the Hubbard or $t$-$J$ model phase diagram that exhibits superconductivity while maintaining analytical control. Although this can be done for the weakly-interacting Hubbard model [6,7], in the limit of strong Hubbard $U$, which corresponds to small $J$ in the $t$-$J$ model, there is evidence that superconductivity does not occur [8–11]. To gain insight on the strongly-interacting regime, the large $J$ limit of the $t$-$J$ model has been studied; but this regime has been shown to be dominated by (unphysical[1]) phase separation [9]. To make progress, many works have considered a large variety of modifications to the $t$-$J$ model in order to improve analytical tractability. Such modifications include explicit symmetry breaking [12,13], large spatial dimension [14], large $N$ [15], nonlocality [16,17], SYK-like nonlocality with large $N$ [18], and replacing the Heisenberg interaction $J$ with an Ising interaction $J_z$ [19–24].

In this work, our goal will be to study the simplest modification to the $t$-$J$ model (that does not enlarge the Hilbert space) such that a superconducting phases exists and can be well-understood with analytical control. Since the nearest-neighbor hopping frustrates the antiferromagnetic order in the $t$-$J$ model, we replace the nearest-neighbor hopping $t$ with a next-nearest-neighbor hopping $t'$ which does not compete with antiferromagnetism. To further simplify, we replace the Heisenberg interaction $J$ with an antiferromagnetic Ising interaction $J_z$.[2] We also add a nearest-neighbor repulsion $V$ to prevent unphysical charge separation. See Fig. 1.

The absence of a nearest-neighbor hopping may be an unrealistic aspect of our model. However, this omission is loosely motivated since nearest-neighbor hopping is strongly suppressed in $t$-$J$-like models near half-filling when $J$ is large [27–29]. Also note that next-nearest-neighbor hopping keeps the fermions on the same sublattice, which is a constraint that can also occur for polarons in an antiferromagnet [30–32]. Thus, our model could also be considered to be a toy model for polarons in an Ising antiferromagnet.

In Sec. 2, we show that in a certain large $J_z$ and $V$ limit, the ground state of the $t'$-$J_z$-$V$ model [Eq. (1)] is antiferromagnetic and the low-energy Hamiltonian can be exactly mapped to a Hamiltonian [Eq. (6)] where the only interaction is an attractive interaction. When the effective attraction is weak, the simplified model can be studied using BCS mean-field theory, which we carry out in detail.

---

[1] Here, phase separation means that a fraction of the system is completely unfilled while the rest is full of electrons. This state is unphysical because it has an infinite energy density when the $1/r$ Coulomb repulsion is not ignored.

[2] The $t$-$t'$-$t''$-$J_z$ model has been studied in Ref. [25,26].

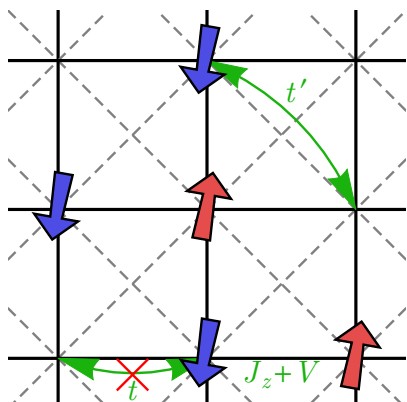

Figure 1: A depiction of the $t'$-$J_z$-$V$ model [Eq. (1)] that we study. This model includes a next-nearest-neighbor hopping $t'$ across the dashed gray links instead of a nearest-neighbor hopping $t$ across the solid black links. The model also includes an antiferromagnetic Ising interaction $J_z$ and nearest-neighbor repulsion $V$ across each solid black link. Unlike a nearest-neighbor hopping $t$, the next-nearest-neighbor hopping $t'$ does not frustrate the antiferromagnetic interaction. The red and blue arrows denote spin up and spin down fermions.

## 2 $t'$-$J_z$-$V$ Model

In this work, we study the $t'$-$J_z$-$V$ model on a square lattice (Fig. 1), which has the following Hamiltonian:

$$H_{t'\text{-}J_z\text{-}V} = t' \sum_{\langle\langle ij\rangle\rangle} \sum_{s=\uparrow,\downarrow} \mathcal{P}\left(c_{is}^{\dagger}c_{js} + c_{js}^{\dagger}c_{is}\right)\mathcal{P} + J_z \sum_{\langle ij\rangle} S_i^z S_j^z + V \sum_{\langle ij\rangle} n_i n_j, \tag{1}$$

with the single-occupancy constraint $n_i = n_{i\uparrow} + n_{i\downarrow} \leq 1$. The first term hops electrons diagonally between next-nearest-neighbor sites $\langle\langle ij\rangle\rangle$ while imposing the $n_i \leq 1$ constraint via the projection operator $\mathcal{P}$, which projects out $n_i = 2$ states. The second term is a nearest-neighbor antiferromagnetic Ising interaction where $S_i^z = \frac{1}{2}(n_{i\uparrow} - n_{i\downarrow})$. The third term is a nearest-neighbor repulsive interaction. We study $H_{t'\text{-}J_z\text{-}V}$ on a square lattice; however, many of our results readily generalize to any bipartite lattice. The model has a $U(1)^4$ symmetry resulting from conserved charge and $z$-component of spin on each sublattice.

It is convenient to redefine the nearest-neighbor repulsion as $V = \frac{1}{4}J_z - V_0$ and rewrite the Hamiltonian as:

$$H_{t'\text{-}J_z\text{-}V_0} = t' \sum_{\langle\langle ij\rangle\rangle,s} \mathcal{P}\left(c_{is}^{\dagger}c_{js} + c_{js}^{\dagger}c_{is}\right)\mathcal{P} + J_z \sum_{\langle ij\rangle}\left(S_i^z S_j^z + \tfrac{1}{4}n_i n_j\right) - V_0 \sum_{\langle ij\rangle} n_i n_j. \tag{2}$$

We will focus on the following limit:

$$V_0 \ll |t'| \ll J_z, \tag{3}$$

with electron filling $\langle n\rangle < 1$.

It is useful to consider the energy levels of two nearest-neighbor sites in the $t' = 0$ limit:

| state | $t' = 0$ energy |
|:---:|:---:|
| $\uparrow\uparrow, \downarrow\downarrow$ | $J_z/2 - v$ |
| $\uparrow 0, \downarrow 0, 0\uparrow, 0\downarrow$ | 0 |
| 00 | 0 |
| $\uparrow\downarrow, \downarrow\uparrow$ | $-v$ |

$$(4)$$

In the above table, ↑ and ↓ refer to spin up and down electrons, while 0 refers to an empty site.

Thus, in the large $J_z$ limit, parallel spins are strongly suppressed. We argue that the ground state never has parallel spins in the $V_0 \ll |t'| \ll J_z$ limit for sufficiently large electron fillings. This occurs because all of the eigenstates have definite $S^z$ spin on each sublattice, and the lowest energy state is a fully-polarized antiferromagnet where one sublattice has only spin-up electrons and the other has only spin-down. This is the lowest-energy symmetry sector since it minimizes the energy from the $J_z$ term and also minimizes the energy of the $t'$ term by allowing for the most electron hopping. See Fig. 1 for an example of a state in this symmetry sector. In Appendix A, we provide a rigorous numerical argument that the ground state is fully antiferromagnetic when $V_0 \ll |t'| \ll J_z$ and for sufficiently large electron filling: $\langle n \rangle > n_c$ where we bound $n_c < 0.265$.

## 2.1 Effective Model

Since the ground states are fully-polarized Ising antiferromagnets, let us consider the antiferromagnetic ground state where the A and B sublattices have only spin-up and spin-down electrions, respectively. It is then convenient to define new electron operators:

$$d_i = \begin{cases} c_{i\uparrow} & i \in A \\ c_{i\downarrow} & j \in B \end{cases}. \tag{5}$$

Within this subspace of only fully-polarized antiferromagnetic states, the $t'$-$J_z$-$V_0$ model [Eq. (2)] simplifies significantly:

$$H_{\text{AF}} = t' \sum_{\langle\langle ij \rangle\rangle} \left( d_i^\dagger d_j + d_j^\dagger d_i \right) - V_0 \sum_{\langle ij \rangle} n_i n_j. \tag{6}$$

That is, the ground states of the $t'$-$J_z$-$V_0$ model can be described by the above Hamiltonian, $H_{\text{AF}}$, which only involves fermions with a next-nearest-neighbor hopping $t'$ and attractive interaction $V_0$.

When $V_0 \ll t'$, we can apply BCS mean-field-theory to study $H_{\text{AF}}$, which we work out in detail in Appendix B. The BCS order parameter is

$$\Delta_\delta = V_0 \langle d_i d_{i+\delta} \rangle, \text{ where } i \in A, \tag{7}$$

where $\delta = \hat{x}, \hat{y}$. The symmetry of the order parameter can be s-wave ($\Delta_x = \Delta_y$) or d-wave ($\Delta_x = -\Delta_y$), depending on the electron filling and sign of $t'$. Since the order parameter $\Delta_\delta$ is not on-site, its Fourier transformation [$\Delta_k$ in Eq. (18)] has nodal lines (where $\Delta_k = 0$) in $k$-space for both s-wave and d-wave symmetry. However, if $\langle n \rangle \neq 1/2$, then the nodal lines never touch the fermi surface for either s-wave and d-wave symmetry, as shown in Fig. 2.

In the $V_0 \ll t'$ limit, the BCS order parameter satisfies the standard BCS gap equation

$$|\Delta_x| = |\Delta_y| = 2\omega e^{-1/V_0 g_0}, \tag{8}$$

where $\omega$ and $g_0$ are parameters, which we calculated numerically and show in Fig. 3 as a function of the filling fraction $\langle n \rangle$.

Although the density of states at the Fermi surface diverges at half filling, $g_0$ and the BCS gap $|\Delta_x|$ (in the weak interaction limit $V_0 \ll t'$) actually decrease as half filling is approached. This might conflict with one's intuition that a large density of states strengthens superconductivity. However, the diverging density of states occurs due to saddle points in the energy dispersion at momenta $k = (\pm\frac{\pi}{2}, \pm\frac{\pi}{2})$ (black dots in Fig. 2), and these saddle points sit on the nodal lines of the BCS order parameter $\Delta_k$. Therefore, these states do not contribute to the BCS gap $\Delta_x$. In Appendix B.1, we mathematically confirm this argument.

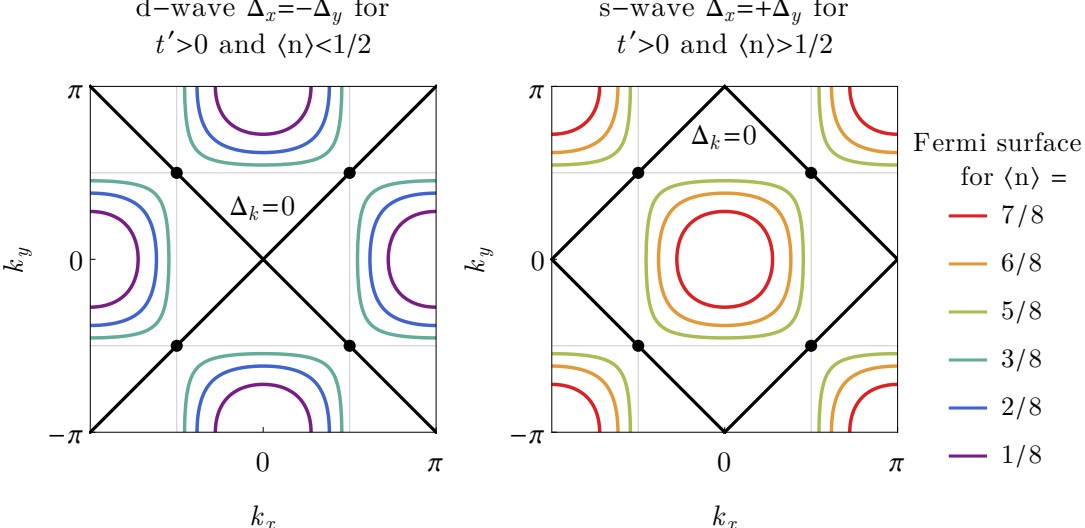

Figure 2: The nodal lines of the order parameter [black lines where $\Delta_k=0$ in Eq. (18)] and the fermi surface for various electron fillings (colored lines). The symmetry of the order parameter depends on the electron filling. When $t' > 0$, the symmetry is d-wave when $\langle n \rangle < 1/2$ and s-wave when $\langle n \rangle > 1/2$. The nodal lines never touch the fermi surface as long as $\langle n \rangle \neq 1/2$. The $t' < 0$ case follows from noting that the physics is symmetric under $t' \to -t'$ and $\langle n \rangle \to 1 - \langle n \rangle$.

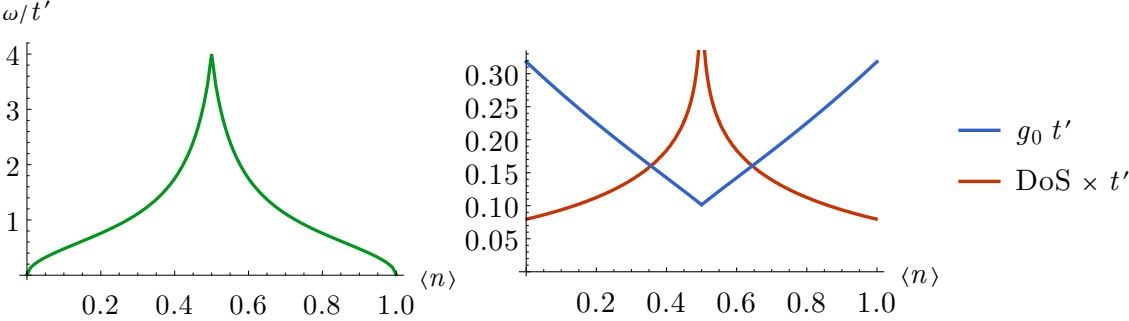

Figure 3: The BCS gap equation parameters $\omega$ (left in green) and $g_0$ (right in blue) from Eq. (8), and the density of states at the Fermi surface (right in red). The parameters are rescaled by $t'$ to make them unitless. The density of states is defined by $\text{DoS} = \int_k \delta(\varepsilon_k)$, where we absorbed the chemical potential $\mu$, which depends on $\langle n \rangle$, into the electron dispersion $\varepsilon_k$ [Eq. (18)]. The density of states has a log divergence at $\langle n \rangle = 1/2$ where $\text{DoS} \sim 0.05 - 0.06 \log |\langle n \rangle - \frac{1}{2}|$.

## 3 Conclusion

As part of a program to identify simple and analytically tractable toy models of superconductivity [33–35], in this work we identify three modifications to the $t$-$J$ model, resulting in the $t'$-$J_z$-$V$ model, that allow for an analytically controlled understanding of its antiferromagnetic d-wave superconducting ground state using BCS theory. Due to the second-nearest-neighbor hopping and antiferromagnetic ground state, the onsite Hubbard repulsion effectively disappears from the effective Hamiltonian $H_{\text{AF}}$ [Eq. (6)], and the antiferromagnetic

Heisenburg term leads to an effective nearest-neighbor attractive interaction $V_0$ in the antiferromagnetic ground state. Since the attractive interaction $V_0$ does not have to compete with an onsite Hubbard interaction (due to its absence in $H_{AF}$), the mechanism for Cooper pairing is very simple and results from a BCS description of the attractive interaction $V_0$. We then studied the small $V_0$ limit in detail using BCS theory. We also discussed why a diverging density of states at the Fermi level does not contribute to superconductivity in our model.

An interesting property of our model is the coexistence of antiferromagnetism and superconductivity. This coexistence has been studied and predicted in a number of works on (sometimes extended) Hubbard and $t$-$J$ models [36–40]. Our model provides an example of such a coexistence in an analytically tractable setting.

It would be interesting to combine the $t$-$J$ and $t'$-$J_z$-$V$ models into a single $t$-$t'$-$J_{xy}$-$J_z$-$V$ model to understand how universal the superconducting state we found is, to what extent it extends into the larger phase diagram, and if it boarders different superconducting states.

In Appendix A we showed that if $V_0 \ll |t'| \ll J_z$ and the electron filling is greater than $n_c = 0.265$, then the ground state is a fully polarized antiferromagnet. However, we did not consider the small filling case $n \ll 1$. It could be possible that sufficiently small fillings also lead to a fully polarized antiferromagnet.

Nevertheless, the $t'$-$J_z$-$V$ model that we studied has a number of limitations. Although it may be applicable to the study of polarons in an Ising antiferromagnet for which a nearest-neighbor hopping is not allowed, the absence of a nearest-neighbor hopping in our model is unnatural for an electron model. Furthermore, the tractable limit of our model was in a large antiferromagnetic interaction $J_z$ limit, which may not be experimentally accessible. Finally, the superconducting state that we found has large Cooper pairs (since it's described by BCS theory) and no gapless nodes (i.e. the lines where the order parameter is zero $\Delta_k = 0$ never touch the Fermi surface). This makes the superconducting state we found qualitatively different from more interesting superconducting states, such as the ones found in the cuprate superconductors [1–4] . In the future, it would be interesting to identify other simple and analytically tractable models with less of these shortcomings, or to include more exotic physics, such as emergent gauge fields [41, 42], while retaining analytical control.

## Acknowledgements

We thank Patrick Lee and Assa Auerbach for helpful discussions.

**Funding information** KS acknowledges support from the Walter Burke Institute for Theoretical Physics at Caltech.

## A  Saturated Antiferromagnetism

In order to reduce the $t'$-$J_z$-$V_0$ model [Eq. (2)] to the effective antiferromagnetic model [Eq. (6)], we need to show that the ground states of the $t'$-$J_z$-$V_0$ model have only spin up electrons on a one sublattice and only spin down electrons on the other sublattice. In this appendix, we argue that this is the case when

$$V_0 \ll |t'| \ll J_z \text{ and } \langle n \rangle \gtrsim 0.265. \tag{9}$$

To show this, we show that Eq. (9) implies that the lowest-energy fully-polarized antiferromagnetic state has a lower energy than any state state with a single flipped spin. By "flipped spin," we mean an electron with a spin in the opposite direction from the

antiferromagnetic order parameter. We expect that if a single spin flip costs energy, then flipping more spins will not result in a lower energy. If this expectation is true, then we have shown that the ground state is a fully polarized antiferromagnet when Eq. (9) is satisfied.

More precisely, assuming $V_0 \ll |t'| \ll J_z$, we numerically calculated a lower bound on the energy cost $E^{\text{flip}}(N)$ to flip a single electron spin for a state with $N$ electrons on a square lattice with $N_{\text{sites}}$ sites. Mathematically, $E^{\text{flip}}(N)$ is defined as

$$
\begin{aligned}
E^{\text{flip}}(N) &= E^{\text{AF}}_{N-1,1} - E^{\text{AF}}_{N,0}, \\
E^{\text{AF}}_{N_1,N_2} &= E\Big(N^{\text{tot}}_{A\uparrow} + N^{\text{tot}}_{B\downarrow} = N_1 \, ; \, N^{\text{tot}}_{A\downarrow} + N^{\text{tot}}_{B\uparrow} = N_2\Big),
\end{aligned}
\tag{10}
$$

where $E^{\text{AF}}_{N_1,N_2}$ is the lowest energy state with $N_1$ electrons that are either spin-up on the A sublattice or spin-down on the B sublattice and $N_2$ electrons that are either spin-down on the A sublattice or spin-up on the B sublattice.

We want to show that $E^{\text{flip}}(N)$ is positive for sufficiently large $\langle n \rangle = N/N_{\text{sites}}$ (as $N \to \infty$). Since we're assuming $V_0 \ll |t'| \ll J_z$, and the $t'$ and $J_z$ terms are sufficient to eliminate any extensive degeneracy, it's sufficient to ignore the attractive $V_0$ term and only consider states in the ground state of the $J_z$ term in Eq. (2). That is, we can simplify the calculation by considering the following limit:

$$
V_0 = 0, \qquad\qquad\qquad J_z = \infty.
\tag{11}
$$

$E^{\text{AF}}_{N,0}$ is the fully-polarized antiferromagnet ground state energy. $E^{\text{AF}}_{N,0}$ can be efficiently calculated since it only involves free fermions since the $J_z$ term does not contribute to fully-polarized states and we are ignoring the $V_0$ term.

$E^{\text{AF}}_{N-1,1}$ is more complicated to calculate, but we can place a lower bound on it. Let $|\Psi^{\text{AF}}_{N-1,1}\rangle$ be an eigenstate with energy $E^{\text{AF}}_{N-1,1}$. Let us decompose $|\Psi^{\text{AF}}_{N-1,1}\rangle$ as a sum of states with a definite position for the flipped spin:

$$
|\Psi^{\text{AF}}_{N-1,1}\rangle = \sqrt{\frac{2}{N_{\text{sites}}}} \sum_{i \in A} \alpha_i c^\dagger_{i\downarrow} |\psi^{(i)}_{N-1,0}\rangle.
\tag{12}
$$

$|\psi^{(i)}_{N-1,0}\rangle$ is a state with $N$ electrons that are either spin-up on the A sublattice or spin-down on the B sublattice, and where $|\psi^{(i)}_{N-1,0}\rangle$ depends on the lattice site $i$ of the flipped spin. Translation symmetry implies that $\alpha_i$ is only a phase (i.e. $|\alpha_i| = 1$) and the states $|\psi^{(i)}_{N-1,0}\rangle$ are related by translation (i.e. $T_\delta |\psi^{(i)}_{N-1,0}\rangle = |\psi^{(i+\delta)}_{N-1,0}\rangle$ where $T_\delta$ is a translation operator).

We can now derive the following bound:

$$
\begin{aligned}
E^{\text{AF}}_{N-1,1} &= \Big\langle \Psi^{\text{AF}}_{N-1,1} \Big| H_{t'\text{-}J_z\text{-}V_0} \Big| \Psi^{\text{AF}}_{N-1,1} \Big\rangle \\
&= \frac{2}{N_{\text{sites}}} \Bigg[ \sum_{\substack{ij \in A \\ i \neq j}} \alpha^*_i \alpha_j \Big\langle \psi^{(i)}_{N-1,0} \Big| c_{i\downarrow} H_{t'} c^\dagger_{j\downarrow} \Big| \psi^{(j)}_{N-1,0} \Big\rangle + \sum_{i \in A} \Big\langle \psi^{(i)}_{N-1,0} \Big| H_{t'} \Big| \psi^{(i)}_{N-1,0} \Big\rangle \Bigg] \\
&= \sum_{j = i \pm \hat{x} \pm \hat{y}} t' \alpha^*_i \alpha_j \Big\langle \psi^{(i)}_{N-1,0} \Big| \psi^{(j)}_{N-1,0} \Big\rangle + \Big\langle \psi^{(i)}_{N-1,0} \Big| H_{t'} \Big| \psi^{(i)}_{N-1,0} \Big\rangle, \text{ where } i \in A \quad (13) \\
&\geq -4|t'| + \widetilde{E}^{\text{AF}}_{N-1,0}, \quad\quad\quad\quad\quad\quad\quad\quad\quad\quad\quad\quad\quad\quad\quad\quad\quad\quad\quad\quad\quad (14)
\end{aligned}
$$

$H_{t'}$ is the $t'$ term in $H_{t'\text{-}J_z\text{-}V_0}$ [Eq. (2)]. Only the $t'$ term contributes due to the $V_0 = 0$ limit [Eq. (11)]. In Eq. (13), $i$ can be any site in the A sublattice. The first term in Eq. (14) results from bounding $t' \alpha^*_i \alpha_j \langle \psi^{(i)}_{N-1,0} | \psi^{(j)}_{N-1,0} \rangle \geq -|t'|$. $\widetilde{E}^{\text{AF}}_{N-1,0}$ is the energy defined in Fig. 4. $\widetilde{E}^{\text{AF}}_{N-1,0}$ bounds the second term in Eq. (13) since it is the ground state energy of $H_{t'}$ subject to the

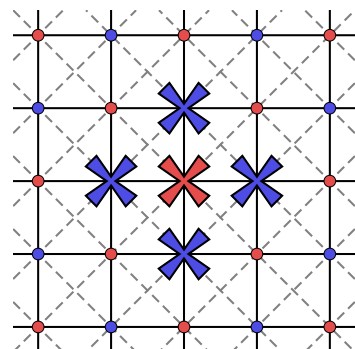

Figure 4: $\widetilde{E}^{\text{AF}}_{N-1,0}$ is the ground state energy of $H_{t'}$ [i.e. the $t'$ term in $H_{t'\text{-}J_z\text{-}V_0}$ from Eq. (2)] with $N-1$ electrons that are either spin-up on the A sublattice (red sites) or spin-down on the B sublattice (blue sites) and subject to the constraint that there are no fermions on the five sites marked with crosses.

same constraint that is enforced upon $|\psi^{(i)}_{N-1,0}\rangle$ [due to $J_z = \infty$ in Eq. (11)]. $\widetilde{E}^{\text{AF}}_{N-1,0}$ can be efficiently calculated since the projection operators $\mathcal{P}$ in $H_{t'}$ act as the identity operator for the electron filling under consideration; thus we only need to calculate the ground state energy of a free fermion Hamiltonian.

In Fig. 5, we plot

$$E^{\text{flip}}(N) \geq -4t' + \widetilde{E}^{\text{AF}}_{N-1,0} - E^{\text{AF}}_{N,0}, \tag{15}$$

where the bound follows from Eqs. (10) and (14). Fig. 5 is therefore evidence that the ground state is a fully-polarized antiferromagnet.

## B  Mean Field Theory

In this appendix, we study $H_{\text{AF}}$ [Eq. (6)] using BCS mean-field theory. We primarily do this to check the symmetry of the BCS order parameter (see e.g. Fig. 2). We also numerically calculate the scaling of the order parameter for weak interactions and display the result in Fig. 3.

We begin with the following BCS mean-field expansion

$$n_i n_j \approx \langle d_i d_j \rangle d_j^\dagger d_i^\dagger + \langle d_i d_j \rangle^* d_i d_j - |\langle d_i d_j \rangle|^2, \tag{16}$$

where we have dropped $O\big(d_i d_j - \langle d_i d_j \rangle\big)^2$ terms.

After applying the above mean-field expansion and a Fourier transformation ($d_k = N^{-1/2} \sum_j e^{-ik \cdot j} d_j$), $H_{\text{AF}}$ becomes

$$H_{\text{BCS}} = \sum_{k}^{0 \leq k_x < \pi} \begin{pmatrix} d_k^\dagger \\ d_{\pi-k} \end{pmatrix} \begin{pmatrix} +\varepsilon_k & -\Delta_k \\ -\Delta_k^* & -\varepsilon_k \end{pmatrix} \begin{pmatrix} d_k \\ d_{\pi-k}^\dagger \end{pmatrix} + \frac{N}{V} \big( |\Delta_x|^2 + |\Delta_y|^2 \big), \tag{17}$$

where we have dropped a constant that does not depend on $\Delta_x$ or $\Delta_y$. The electron dispersion $\varepsilon_k$ and gap function $\Delta_k$ are:

$$\begin{aligned} \varepsilon_k &= 4t' \cos k_x \cos k_y - \mu, \\ \Delta_k &= 2\Delta_x \cos k_x + 2\Delta_y \cos k_y, \end{aligned} \tag{18}$$

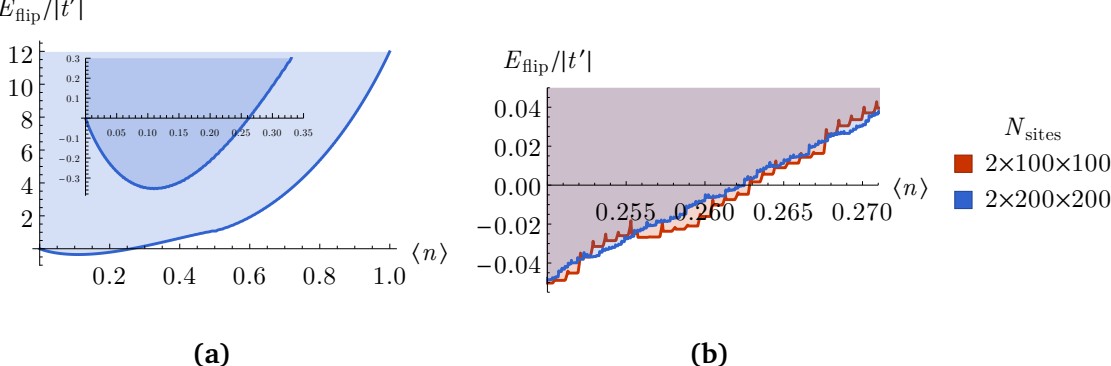

**(a)** **(b)**

Figure 5: A lower bound on the energy $E^{\text{flip}}$ [Eq. (10)] required to flip an electron spin on one of the sublattices when $V_0 \ll |t'| \ll J_z$. We used a square lattice with $N_{\text{sites}} = 2 \times 200 \times 200$, where the A and B sublattices are each $200 \times 200$ square lattices with periodic boundary conditions which are rotated $45°$ with respect to Fig. 1. We expect the reduced model [Eq. (6)] to be valid when $E^{\text{flip}} > 0$. The figure shows that there is a critical filling $n_c$ such that $E^{\text{flip}} > 0$ for all $\langle n \rangle > n_c$. **(b)** Zooming in suggests an upper bound on the critical filling: $n_c < 0.265$. We also show the $N_{\text{sites}} = 2 \times 100 \times 100$ lattice result as evidence that larger system sizes would only improve our bound.

where $\mu$ is the chemical potential. The mean-field order parameter $\Delta_\delta$ is defined by

$$\Delta_\delta = V_0 \langle d_i d_{i+\delta} \rangle, \text{ where } i \in A. \tag{19}$$

$\sum_k^{0 \leq k_x < \pi}$ sums over all momenta $k$ in the half-Brillouin zone with $0 \leq k_x < \pi$. $\pi - k$ is defined by $\pi - k = (\pi - k_x, \pi - k_y)$ where $k = (k_x, k_y)$.

$H_{\text{BCS}}$ can be diagonalized by a Bogoliubov transformation:

$$H_{\text{Bogoliubov}} = \sum_k^{0 \leq k_x < \pi} \begin{pmatrix} \alpha_k^\dagger \\ \beta_k^\dagger \end{pmatrix} \begin{pmatrix} +\lambda_k & 0 \\ 0 & -\lambda_k \end{pmatrix} \begin{pmatrix} \alpha_k \\ \beta_k \end{pmatrix} + \frac{N}{V_0} \left( |\Delta_x|^2 + |\Delta_y|^2 \right), \tag{20}$$

$$\lambda_k = \sqrt{\varepsilon_k^2 + |\Delta_k^2|}, \tag{21}$$

where $\pm \lambda_k$ are the Bogoliubov quasi-particle energies. The Bogoliubov quasi-particle operators $\alpha_k$ and $\beta_k$ with $0 \leq k_x < \pi$ are defined in terms of the electron operators $d_k$ by the following Bogoliubov transformation:

$$\begin{pmatrix} d_k \\ d_{\pi-k}^\dagger \end{pmatrix} = \begin{pmatrix} +\cos\theta_k & +\sin\theta_k e^{+i\phi_k} \\ -\sin\theta_k e^{-i\phi_k} & +\cos\theta_k \end{pmatrix} \begin{pmatrix} \alpha_k \\ \beta_k \end{pmatrix}, \tag{22}$$

where the angle $0 < \theta_k < \pi/4$ is defined by $\tan(2\theta_k) = |\Delta_k|/\varepsilon_k$ and $\phi$ is the phase of $\Delta_k = |\Delta_k| e^{i\phi_k}$.

The order parameters $\Delta_x$ and $\Delta_y$ can be obtained by variationally minimizing the ground state energy density

$$\frac{E}{N} = -\frac{1}{2} \int_k \lambda_k + V_0^{-1} \left( |\Delta_x|^2 + |\Delta_y|^2 \right), \tag{23}$$

or by solving the self-consistency condition:

$$\Delta_\delta = V_0 \langle d_i d_{i+\delta} \rangle, \text{ where } i \in A$$
$$= \frac{V_0}{2N} \int_k \cos k_\delta \frac{\Delta_k}{\lambda_k}, \tag{24}$$

where $\int_k = \int \frac{dk_x}{2\pi} \frac{dk_y}{2\pi}$.

We will assume that $\Delta_x$ and $\Delta_y$ are related by a phase $s$ ($|s| = 1$):

$$\Delta_y = s\Delta_x. \tag{25}$$

Solving the self-consistency Eq. (24) for $V_0^{-1}$ results in

$$V_0^{-1} = \frac{1}{2} \int_k \left| \cos k_x + s \cos k_y \right|^2 / \lambda_k. \tag{26}$$

From the above Eq. (26), we can calculate the interaction strength $V_0$ as a function of the order parameter $\Delta_y = s\Delta_x$ and chemical potential $\mu$. We use Eq. (23) to find which order parameter symmetry ($s = 1$, i, or $-1$) gives the lowest ground state energy; the result in summarized in Fig. 2.

We numerically calculate the scaling coefficients of the order parameter in the weak interaction limit $V_0 \ll |t'|$ by calculating $V_0$ from Eq. (26) for a few very small values of the order parameter: $|\Delta_x/t'| \sim 10^{-5}$. We then fit the resulting $(V_0, |\Delta_x|)$ data to the standard BCS gap equation

$$|\Delta_x| = |\Delta_y| = 2\omega e^{-1/V_0 g_0} \tag{27}$$

using $\omega$ and $g_0$ as free parameters. The result is shown in Fig. 3 as a function of the filling fraction $\langle n \rangle$.

## B.1 Approximate Gap Scaling

In the usual s-wave BCS theory, $g_0$ in Eq. (8) is approximately equal to the density of states at the Fermi level [43]. However, Fig. 3 shows that this is clearly not the case in our model. This occurs because in Eq. (26), the integral can not be reformulated in terms of the density of states $g(\varepsilon)$ as just a function of the energy $\varepsilon$. Rather, one requires a density of states $g(\varepsilon, \chi)$ that is also a function of the shape of the gap function $\chi = |\Delta_k/\Delta_x|$, which can be seen by rewriting Eq. (26) as

$$V_0^{-1} = \frac{1}{8} \int d\varepsilon \int d\chi \, g(\varepsilon, \chi) \frac{\chi^2}{\sqrt{\varepsilon^2 + |\Delta_x^2| \chi^2}}, \tag{28}$$

$$g(\varepsilon, \chi) = \int_k \delta(\varepsilon_k - \varepsilon) \delta(|\Delta_k/\Delta_x| - \chi). \tag{29}$$

In Fig. 6, we plot $g(\varepsilon, \chi)$ for our model.

Note that the integral in Eq. (28) is dominated by the region near the Fermi level where $\varepsilon = 0$. Thus, similar to ordinary BCS theory, we can approximate the $\varepsilon$ dependence as a box distribution:

$$g(\varepsilon, \chi) \approx \begin{cases} g(\chi) & |\varepsilon| < W \\ 0 & \text{otherwise} \end{cases}. \tag{30}$$

We can now perform the $\varepsilon$ integral in Eq. (28) to obtain:

$$V_0^{-1} = \int d\chi \, \frac{1}{4} g(\chi) \chi^2 \ln \frac{2W}{|\Delta_x| \chi}. \tag{31}$$

Solving the above equation for $\Delta_x$ results in the BCS gap equation [Eq. (27)] with the following

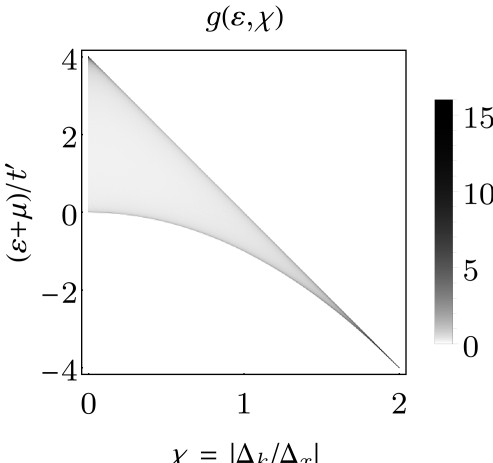

Figure 6: The density of states $g(\varepsilon, \chi)$ [Eq. (29)] as a function of the single-particle energy $\varepsilon$ and gap function $\chi$ for our model [Eq. (18)] when $\Delta_x = -\Delta_y$ (which occurs when $\mu/t' < 0$). The $\Delta_x = +\Delta_y$ case is obtained by reflecting $\varepsilon + \mu \to -\varepsilon - \mu$. The density of states $g(\varepsilon)$ as a function of only the single-particle energy $\varepsilon$ is shown in Fig. 3. The grayscale legend should not be taken seriously at the bottom-right corner where $g(\varepsilon, \chi)$ diverges.

BCS parameters:

$$
\begin{aligned}
g_0 &= \int \mathrm{d}\chi \, \frac{1}{4} g(\chi) \chi^2, \\
\omega &= W \exp\left(-\frac{1}{g_0} \int \mathrm{d}\chi \, \frac{1}{4} g(\chi) \chi^2 \ln \chi\right) \\
&= W \exp\left(-\langle \ln \chi \rangle_{P(\chi) = \frac{1}{4g_0} g(\chi) \chi^2}\right).
\end{aligned}
\tag{32}
$$

$g_0$ does not depend on $W$, which shows that $g_0$ only depends on the states near the Fermi level $\varepsilon = 0$. We also see that states where the gap function $\chi = |\Delta_k/\Delta_x|$ is larger contribute the most to $g_0$. In particular, states along the nodal lines of $\Delta_k$ (i.e. where $\chi = \Delta_k = 0$) do not contribute to $g_0$. This mathematically explains our intuition for $g_0$ that we explained in the last paragraph of Sec. 2.1.

$\omega$ does depend on $W$, and therefore $\omega$ also depends on the states away from the Fermi level $\varepsilon = 0$. $\omega$ is most intuitively expressed in terms of an expectation value of $\langle \ln \chi \rangle$ where $\chi$ is thought of as a random variable with the probability distribution $P(\chi) = \frac{1}{4g_0} g(\chi) \chi^2$.

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
