# Peer review of "Unfrustrating the t-J Model: Exact d-wave BCS Superconductivity in the t'-J_z-V Model"

_SciPost Physics, doi:SciPost Phys. 7, 046 (2019)_

## Round 1 · Referee Report · Anonymous · 2019-9-29

Report

This work introduces a new, physically motivated, member of the t-J-V type models and solves it. In the analytical solution of this system, superconductivity appears naturally along with antiferromagnetism. Very few models display the ease with which this is derived here. I think that it is very refreshing to see a new model with such controlled results.

Although the title contains the word "exact", the calculations invoke the (standard BCS) mean-field approximation (Appendix B) as well as analysis (in Appendix A) in various limit which enable analytical results. I think that it would be better to replace that word by something else.

As the author wrote, a possible weakness might be that only next-nearest-neighbor hopping is allowed. However, that is indeed very well physically motivated noting early works. Indeed, the dispersion relation for a single hole in an antiferromagnetic background may indeed display only harmonics associated with next nearest neighbor hopping. An illuminating work where this is further made lucid is

E. Louis, F. Guinea, M. P. L´opez Sancho, and J. A. Verg´es, Phys. Rev. B 59, 14005 (1999).

This dominant nearest neighbor hopping in an antiferromagnetic background was employed, in the past, to motivate kinetically driven stripe formation and pairing.

With the removal of the nearest neighbor hopping, the new t'-J_{z}-V model of the author is quite remarkable in the ease in which many results follow.

I strongly recommend the publication of this work with the possible minor change regarding the wording of the title.

---

## Editorial Decision

published